# Morphometric frequency and spectrum of gamma-ray-induced chlorophyll mutants identified by phenotype and development of novel variants in lentil (*Lens culinaris* Medik.)

Biswajit Pramanik[1], Sandip Debnath[1]*, Mehdi Rahimi[2], Md. Mostofa Uddin Helal[3], Rakibul Hasan[4]*

1 Department of Genetics and Plant Breeding, Palli Siksha Bhavana (Institute of Agriculture), Visva-Bharati University, Sriniketan, West Bengal, India, 2 Department of Biotechnology, Institute of Science and High Technology and Environmental Sciences, Graduate University of Advanced Technology, Kerman, Iran, 3 Institute of Wheat Research, State Key Laboratory of Sustainable Dryland Agriculture, Shanxi Agricultural University, Linfen, China, 4 Department of Plant Pathology and Seed Science, Sylhet Agricultural University, Sylhet, Bangladesh

* rakib.ppath@sau.ac.bd (RH); sandip.debnath@visva-bharati.ac.in (SD)

**Data Availability Statement:** The authors confirm that the data supporting the findings of this study are available within the article.

## Abstract

Genetic variations are a crucial source of germplasm heterogeneity, as they contribute to the development of new traits for plant breeding by offering an allele resource. Gamma rays have been widely used as a physical agent to produce mutations in plants, and their mutagenic effect has attracted much attention. Nonetheless, few studies have examined the whole mutation spectrum in large-scale phenotypic evaluations. To comprehensively investigate the mutagenic effects of gamma irradiation on lentils, biological consequences on the $M_1$ generation and substantial phenotypic screening on the $M_2$ generation were undertaken. Additionally, the study followed the selected mutants into the $M_3$ generation to evaluate the agronomic traits of interest for crop improvement. Seeds of lentil variety *Moitree* were irradiated with a range of acute gamma irradiation doses (0, 100, 150, 200, 250, 300, and 350 Gy) to induce unique genetic variability. This research focused on determining the $GR_{50}$ value while considering seedling parameters and examining the status of pollen fertility while comparing the effects of the gamma irradiation dosages. The $GR_{50}$ value was determined to be 217.2 Gy using the seedling parameters. Pollens from untreated seed-grown plants were approximately 85% fertile, but those treated with the maximum dosage (350 Gy) were approximately 28% fertile. Numerous chlorophyll and morphological mutants were produced in the $M_2$ generation, with the 300 Gy -treated seeds being the most abundant, followed by the 250 Gy -treated seeds. This demonstrated that an appropriate dosage of gamma rays was advantageous when seeking to generate elite germplasm resources for one or multiple traits. Selected mutants in the $M_3$ generation showed improved agronomic traits, including plant height, root length, number of pods per plant, and yield per plant. These investigations will contribute to a comprehensive understanding of the mutagenic effects and actions of gamma rays, providing a basis for the selection and design of suitable mutagens. This will facilitate the development of more controlled mutagenesis protocols for

**Funding:** The author(s) received no specific funding for this work.

**Competing interests:** The authors have declared that no competing interests exist.

plant breeding and help guide future research directions for crop improvement using radiation-induced mutation breeding techniques.

## Introduction

Lentil (*Lens culinaris* Medik.) is an important pulse crop in India. It is a member of the Fabaceae family with chromosome number 2n = 2x = 14 [1]. It is an annual edible legume with lens-shaped pods and purse-shaped seeds [2]. This oldest known legume originated in the Near East [3], from whence it spread to Canada, Turkey, and Southeast Asian nations such as India, China, Nepal, Afghanistan, Pakistan, and Bangladesh, among others [4]. Lentil seeds are a good source of a variety of proteins, vitamins, and minerals that are required for human nutrition [5]. Additionally, the availability of more tryptophan and lysine makes it a good complement to rice or wheat, which is deficient in that specific amino acid and thus offers balanced nutrition in the human diet. Because the legume "lentil" is a high-protein legume (containing approximately 24.1% protein), crop improvement techniques may have a major impact [6]. Surprisingly, socioeconomic and agroecological constraints are the primary impediments to the development of this food legume across India, despite its great nutritional value. Numerous biotic and abiotic stress conditions significantly reduce the yield potential of lentils, particularly in resource-poor areas worldwide [7]. The indigenous cultivars of lentil in India belonged to a specific ecotype (Pilosae), which appears to lack notable variability for phenological, morphological, yield- and resistance-associated traits, resulting in the failure of lentil improvement programmes for a longer duration [8]. The day length bottleneck hampered the transfer of lentil germplasm into the Indo-Gangetic plain, resulting in decreased yield potential in that particular geographical region [9]. Additionally, the small and delicate blooms hampered the success rate of artificial hybridization by up to 20%–50% due to the difficulties of emasculation and pollination, which resulted in injury during these procedures. As a consequence, cross-pollination has become a time-consuming process in lentils [8]. Additionally, the genus *Lens* was notable for the development of crossability barriers during interspecific hybridization within and across species [10]. Eventually, these processes collectively resulted in the lentil's narrow genetic base. Various scholars previously identified this narrow genetic basis as the primary restraint on lentil production [11]. Additionally, adaptive specificity and repeated failures to employ various unproductive foreign germplasms contribute to the failure of lentil genetic progress [12]. Thus, given the parameters outlined above, mutagenesis might be seen as a supplementary and novel breeding strategy. It is capable of generating such a variation that does not exist in the organism's genetic background. As a result, this feature may be used to enhance a plethora of desirable characteristics in a plant species. A wide range of mutation methods have been used, including physical and chemical mutagens such as gamma radiation, X-rays, heavy metal ions, and protons, as well as biotechniques such as genetic modification, transgenic, or gene editing based on CRISPR/Cas9 technology. Many different forms of DNA damage may be caused by photon radiation, including the nucleotide alterations and strand breaks caused by oxidized bases, abasic sites, single-strand breaks, and double-strand breaks that gamma radiation can cause. If this DNA damage is not addressed or is repaired ambiguously, mutations, such as single-base substitutions, deletions, insertions, inversions, or translocations, may arise at the genome scale and eventually lead to alterations in the phenotypic traits of the organisms [13]. Numerous researchers have proposed and implemented mutation breeding as an effective tool for this kind of improvement in a variety of crops [11, 14–22].

Amin, et al. [11] published a report indicating a dose-dependent decrease in the germination of lentil seeds treated with MMS (methyl methanesulfonate). Additionally, they noted that 2% (v/v) DMSO (dimethyl sulfoxide) inhibited MMS action. Gaur, et al. [17] have observed similar results utilizing gamma rays as a mutagen on pigeon pea. Khursheed, et al. [18] evaluated the combined impact of gamma rays and ethyl methanesulfonate (EMS) on the mutagenesis potential of faba bean (*Vicia faba* L.) seeds by determining the germination percentage and seedling height.

As a result, this research was undertaken to find the optimal dosage of acute gamma-irradiated mutagenesis for the improvement of lentil's agronomically significant features to achieve a greater mutation frequency [22, 23]. Additionally, the immediate impact of gamma irradiation on seedling and sterility parameters is discussed in this section. Additionally, the influence of determining the $GR_{50}$ value by the detection of a large number of mutants, both chlorophyll and morphological, was shown in this study. These studies focused on the lentil variety "Moitree." It is a well-known variety among farmers in West Bengal, India, as well as a national check because of its considerably greater production performance than other varieties, even when seeded late. As a result, a group of researchers used this variety extensively in their experiments [24–26]. The creation of numerous chlorophyll and morphological mutants in the subsequent generation would be crucial for the identification of new alleles and might potentially be released as varieties or pre-breeding material in the near term if they are found to be economically sustainable and agronomically useful.

Thus, in this study, particular emphasis has been placed on examining seedling traits of the $M_1$ generation. Analyzing these traits is essential, as it is considered the most reliable method to detect mutants in early generations [27]. By focusing on seedling characters in the $M_1$ generation, researchers can effectively identify and isolate favorable mutants, which can be further evaluated for their agronomic performance and potential use in breeding programs [28]. This early detection of mutants allows for a more streamlined approach to crop improvement, ensuring that only the most promising mutant lines are carried forward for further analysis and development [29]. Consequently, the importance of studying seedling traits in the $M_1$ generation is a crucial aspect of this research, contributing to the overall objective of finding the optimal dosage of acute gamma-irradiated mutagenesis for the improvement of agronomically significant traits in subsequent generations of lentils and achieving a higher mutation frequency.

## Materials and methods

### Determination of $GR_{50}$ value

Approximately 35000 homogenous, uniform, dry and healthy seeds of lentil (variety: *Moitree*) were collected from the Department of Genetics and Plant Breeding, Palli Siksha Bhavana (Institute of Agriculture), Visva-Bharati, Sriniketan. Among them, every 1/7th of the total seeds, *i.e.*, almost 5000 seeds were exposed to acute gamma irradiation of seven doses including 0, 100, 150, 200, 250, 300, and 350 Gy, respectively, at the rate of 7 sec per 10 Gy radiation. The irradiation procedure was executed in the gamma chamber (GC-6000) of RNARC, BCKV, West Bengal, India. Formal safety precautions, *i.e.*, maintaining a safe distance of 10 m, using proper gloves and kits, closing the room during irradiation procedures, etc., were appropriately followed. A few irradiated seeds of each dose along with the untreated one (used as control; denoted by the irradiation dose of 0 Gy above) were germinated following the petri-plate method. Each filter paper (size 9), soaked with double-distilled water, was placed on twenty-one petri plates, properly marked with the dosage. Seeds of each dose were grown on those specifically marked Petri plates and incubated at 25±2˚C in the laboratory. Ten seeds of each

irradiation dose and control were grown on Petri plates with three replications. For each dose, germination percentage (%), various seedling parameters *viz.* shoot length (cm), root length (cm), seedling height (cm), seedling vigor index, etc., were recorded for 7 and 14 days, respectively.

## Increase in $M_1$ generation

Excluding the $GR_{50}$ value determining experiment, the rest of the seeds irradiated with six doses of gamma rays (100, 150, 200, 250, 300, and 350 Gy) along with the control (untreated seeds) were sown in the field in the year of 2020–21. A few pollens were collected from the floral buds for the following experiment during the pre-flowering stage. From the rest, a large number of $M_1$ populations were developed from those seeds. A large number of pods as well as seeds were collected from individual plants from the population for raising the next generation, *i.e.*, $M_2$ population.

## Pollen fertility study

Pollens were collected from the floral buds of irradiated seed-grown *in vivo* $M_1$ plants along with controls. After collection, the pollens of each dose and control were treated with 1% (w/v) iodide-potassium iodide (IKI) [30] solution for a few seconds, after which they were minutely observed under the 40X objective lens of a compound light microscope (Olympus OIC) in the laboratory, as mentioned earlier. The pollen fertility percentage was calculated by implementing the following formula:

$$Pollen\ fertility\ percentage\ =\ \frac{Number\ of\ fertile\ pollens\ in\ a\ given\ area}{Number\ of\ total\ pollens\ in\ that\ given\ area} \times 100$$

Pollens of all doses and the control were observed under the microscope with three replications each. Pollens from control plants were compared with those of each dose to check whether there was any reduction in pollen fertility percentage.

## Identification of candidate mutants in the $M_2$ generation

Seeds obtained from the $M_1$ generation were sown in the next generation for the purpose of growing the $M_2$ population in the aforementioned field using a plant-to-progeny method in 2021–22. Row-to-row and plant-to-plant spacings of 30 cm and 15 cm, respectively, were maintained. 122 individual plants were selected across the field based on their chlorophyll content and numerous morphological characteristics to determine the mutagenic effects of the acute gamma irradiation dosages (0, 100, 150, 200, 250, 300, and 350 Gy) administered in the previous generation. After ten days of germination, those candidate mutants were appropriately tagged and recorded. Additionally, a dose-dependent normal plant population was also determined. The observed plant mutants were handled as per the relevant regulations and guidelines of IAEA.

## Selection of $M_3$ mutants with improved agronomic characteristics

The 122 candidate mutants selected in the $M_2$ generation were subjected to the plant-to-progeny method again for raising the $M_3$ population in the same field during the following year (2022–23). Row-to-row and plant-to-plant spacings were maintained at 30 cm and 15 cm, respectively, consistent with the previous generation. Six individual mutants were chosen from four of the 122 families in the current population derived from the $M_2$ generation, based on their enhanced agronomic features such as increased plant height, root length, number of

pods, and yield per plant. The observed plant mutants were managed in accordance with the relevant regulations and guidelines of the IAEA in this generation as well.

### Statistical data analysis

The experiment involving the calculation of the dose of growth reduction was organized in a completely randomized block design with seven levels of gamma irradiation (including control) and three treatments, where the radiation levels were randomly arranged. One-way analysis of variance (ANOVA) for all the parameters studied in the experiments was carried out at a 5% significance level to test whether the observed averages of the treatment levels were significantly different. Probit analysis [31] was executed to determine the $GR_{50}$ value of all the parameters of interest. All statistical analyses were performed using the SPSS (version 20.0, SPSS Inc., Chicago, IL, USA) software package.

## Results

### $GR_{50}$ value

In this study, the germination percentage was highest (100%) in the control seeds, whereas retardation in germination efficiency and germination percentage was found with increasing radiation dose (Figs 1 and 2A). Similarly, the control seedlings showed the best outcome concerning all other seedling parameters, *i.e.*, shoot length, root length, total seedling length, seedling vigor index, etc. (Table 1). The seedling vigor index was calculated according to the formula germination percentage × seedling length for each dose, which also provided a similar graph to seedling height in this study (Fig 2B and Table 1). Based on the total seedling length and root length, the $GR_{50}$ value was measured, which was 217.2 Gy (Fig 3A and 3B).

### Pollen fertility

Similar to the last experiment, the floral buds of the control seedlings were found to carry the maximum number of fertile pollens (almost 85%) (Fig 4A and Table 1). On the other hand, 350 Gy gamma-irradiated pollens showed the lowest fertility (27–29%) among all others (Fig 4G and Table 1). Notably, the pollens also exhibited the fertility percentage in descending order (Fig 4A–4G) when placed with elevating doses of gamma radiation in the graph (Fig 4H).

### Frequency of candidate mutants in the $M_2$ generation

During the $M_2$ generation, potential mutants (Fig 5) were thoroughly identified using agronomic and morphological parameters. Four distinct kinds of chlorophyll mutants, viz., albina (Fig 6B), xantha (Fig 6C), chlorina (Fig 6D), and xantha-viridis (Fig 6E), were identified (Table 2). Additionally, a few leaf morphological (Fig 6I–6T and 6V and 6W) and plant morphological mutants (Fig 6H–6J) were also detected (Table 2).

### Superiority of candidate $M_3$ mutants over control as well as $M_2$ generation mean

Table 3 presents the data of selected $M_3$ generation mutants derived from this mutation breeding experiment. It compares the plant height, root length, number of pods per plant, and yield per plant of the $M_3$ generation mutants with the $M_2$ mean and control values. The table highlights the increments observed in the selected $M_3$ generation mutants compared to both the $M_2$ mean and control values, emphasizing the positive effects of mutation breeding on lentil plant traits. This information can be useful for breeders to choose the best performing mutants for further crop improvement programs. For instance, $VBM_3Sel-34$ showed an 18.5 cm

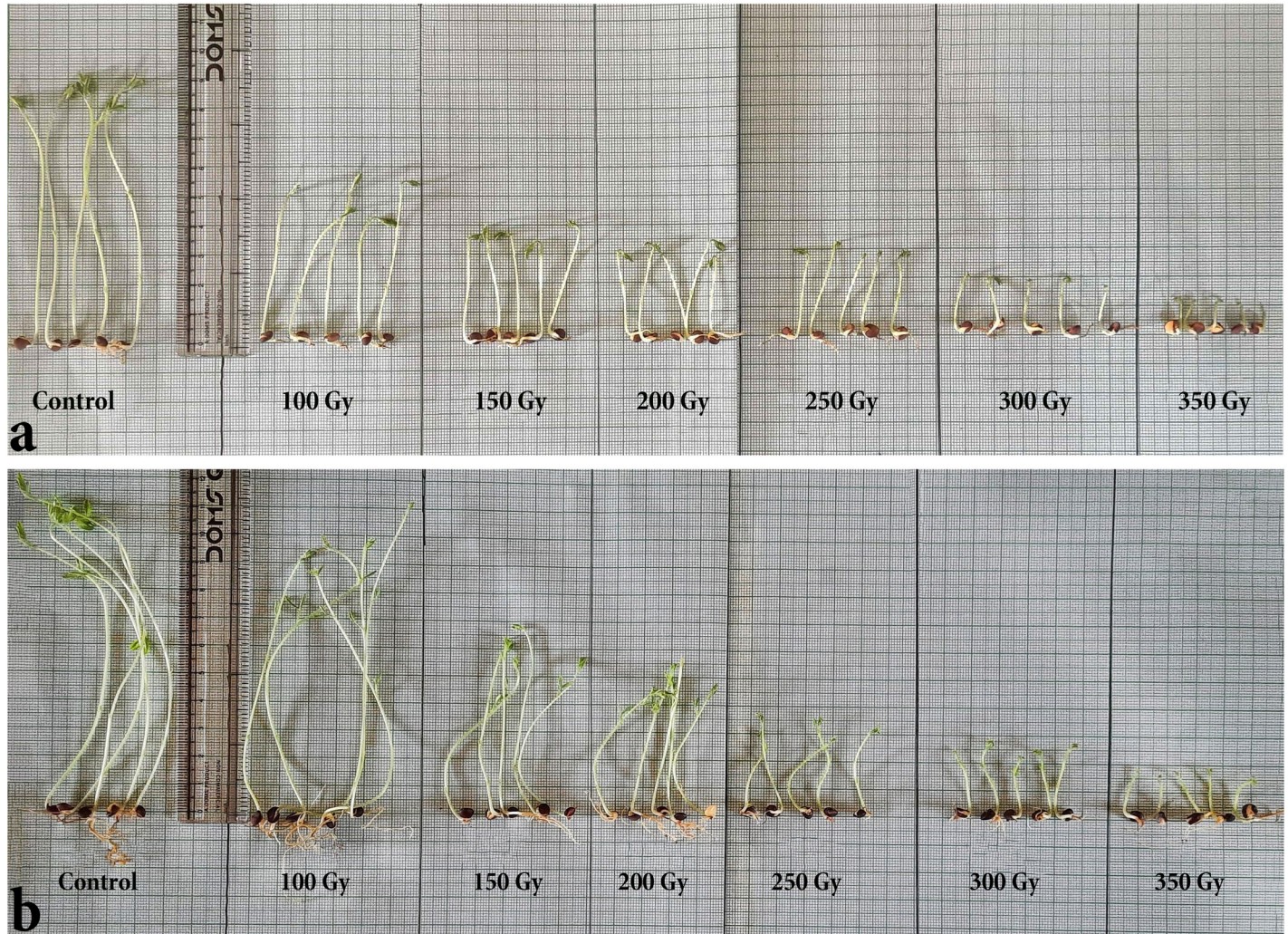

**Fig 1. Immediate effect of different doses of gamma irradiation (0, 100, 150, 200, 250, 300 and 350 Gy) on seedling parameters of lentil (*Lens culinaris* Medik.).**
(a) Growth status of seedlings after 7 days of experiment. (b) Growth status of seedlings after 14 days of experiment.

increase in plant height from the $M_2$ mean and a 26.6 cm increase from the control. This mutant also displayed a 4.7 cm increment in root length from the $M_2$ mean and an 8 cm increment from the control. The number of pods per plant increased by 8 and 14 when compared to the $M_2$ mean and control, respectively. Furthermore, the yield per plant for VBM$_3$Sel-34 increased by 0.55 g from the $M_2$ mean and 1.11 g from the control.

Similar trends can be observed in the other selected $M_3$ generation mutants, such as VBM$_3$Sel-57, VBM$_3$Sel-109, VBM$_3$Sel-178, VBM$_3$Sel-259, and VBM$_3$Sel-357. All these mutants exhibited increments in plant height, root length, number of pods per plant, and yield per plant compared to their respective $M_2$ mean and control values.

This indicates the success of the mutation breeding program in enhancing the agronomic traits of lentil plants, which can ultimately lead to improved crop performance and yield (Fig 7).

Data analysis shows that the selected $M_3$ generation mutants have significantly improved agronomic traits when compared to the $M_2$ mean and control plants. This demonstrates the effectiveness of mutation breeding in enhancing the agronomic traits of lentil plants. The information presented can be valuable for breeders in selecting the best performing mutants

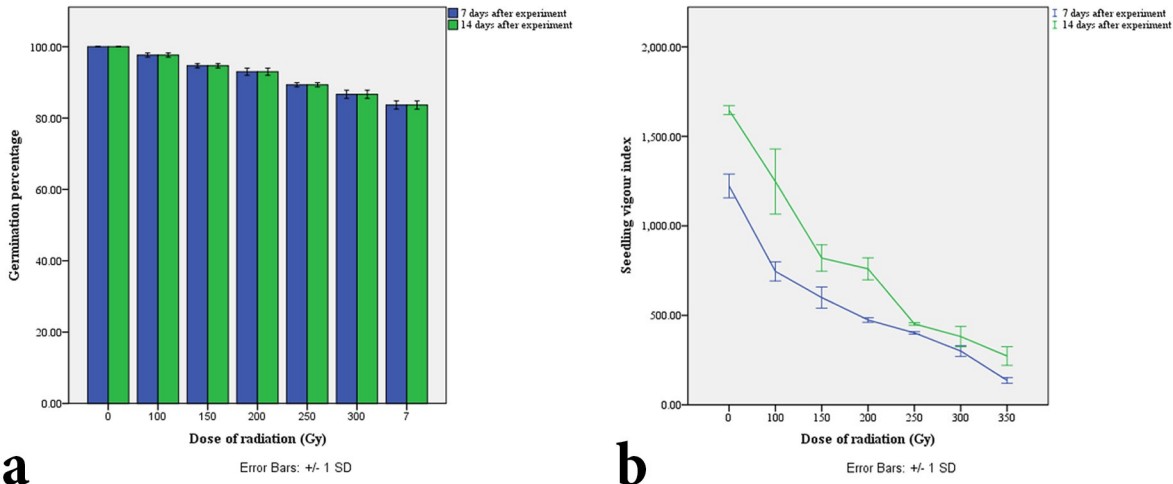

**Fig 2.** Graphical representation of effect of different doses of acute gamma irradiation (0, 100, 150, 200, 250, 300, and 350 Gy) on (a) gemination percentage at 7 and 14 days after experiment, respectively. (b) seedling vigor index at 7 and 14 days after experiment, respectively.

## Discussion

Prior to initiating experiments on colossal scale mutant breeding, $LD_{50}$ and $GR_{50}$ dosages are essential [32]. These $GR_{50}$ and $LD_{50}$ values are crucial because they result in a greater likelihood of acquiring desirable mutants [22]. According to Álvarez-Holguín, et al. [33], lower doses of gamma irradiation result in minor alterations to the genome, but higher doses result in undesired or deadly mutations, *i.e.*, the higher the doses, the greater the mortality. Therefore, the undesirable mutations obtained remain a way away from the expected conclusion

**Table 1. Effect of different doses of gamma radiation on seedling parameters and pollen fertility status in lentils.**

| Dose of radiation | Germination percentage | | Shoot length (cm) | | Root length (cm) | | Seedling length (cm) | | Seedling vigor index | | Pollen fertility percentage |
|---|---|---|---|---|---|---|---|---|---|---|---|
| | 7 DAS | 14 DAS | 7 DAS | 14 DAS | 7 DAS | 14 DAS | 7 DAS | 14 DAS | 7 DAS | 14 DAS | |
| Control (0) | 100 | 100 | 9.06 ±0.06 | 13.6±0.21 | 3.17 ±0.32 | 3.86 ±0.35 | 12.23 ±0.38 | 16.47 ±0.14 | 1223.33 ±38.44 | 1646.67 ±14.53 | 85.27±1.06 |
| 100 | 97.67 ±0.33 | 97.67 ±0.33 | 6.06 ±0.12 | 10.63 ±0.71 | 1.76±0.3 | 2.3±0.26 | 7.63±0.29 | 12.77 ±1.03 | 745.67 ±30.83 | 1247.57 ±104.98 | 69.5±1.85 |
| 150 | 94.67 ±0.33 | 94.67 ±0.33 | 4.6±0.15 | 7.3±0.21 | 1.63 ±0.44 | 1.76 ±0.67 | 6.33±0.38 | 8.67±0.48 | 599.3±34.16 | 820.13±42.57 | 59.43±1.29 |
| 200 | 93±0.57 | 93±0.57 | 3.6±0.05 | 6.87±0.12 | 1.56 ±0.17 | 1.63 ±0.44 | 5.1±0.05 | 8.17±0.34 | 474.33±7.33 | 759.73±35.5 | 52.16±1.64 |
| 250 | 89.33 ±0.33 | 89.33 ±0.33 | 2.73 ±0.23 | 4.8±0.05 | 1.5±0.11 | 1.56 ±0.08 | 4.5±0.05 | 5.07±0.03 | 401.97±3.94 | 452.63±4.09 | 45.2±1.39 |
| 300 | 86.67 ±0.67 | 86.67 ±0.67 | 2±0.11 | 2.96±0.06 | 1.46 ±0.26 | 1.43 ±0.44 | 3.47±0.17 | 4.4±0.37 | 300.67 ±17.58 | 381.4±33.08 | 36.76±2.11 |
| 350 | 83.67 ±0.67 | 83.67 ±0.67 | 0.96 ±0.03 | 1.97±0.03 | 0.66 ±0.08 | 1.3±0.35 | 1.63±0.12 | 3.27±0.38 | 136.5±9.07 | 272.8±30.24 | 28.1±1.53 |

[1] Tables may have a footer.

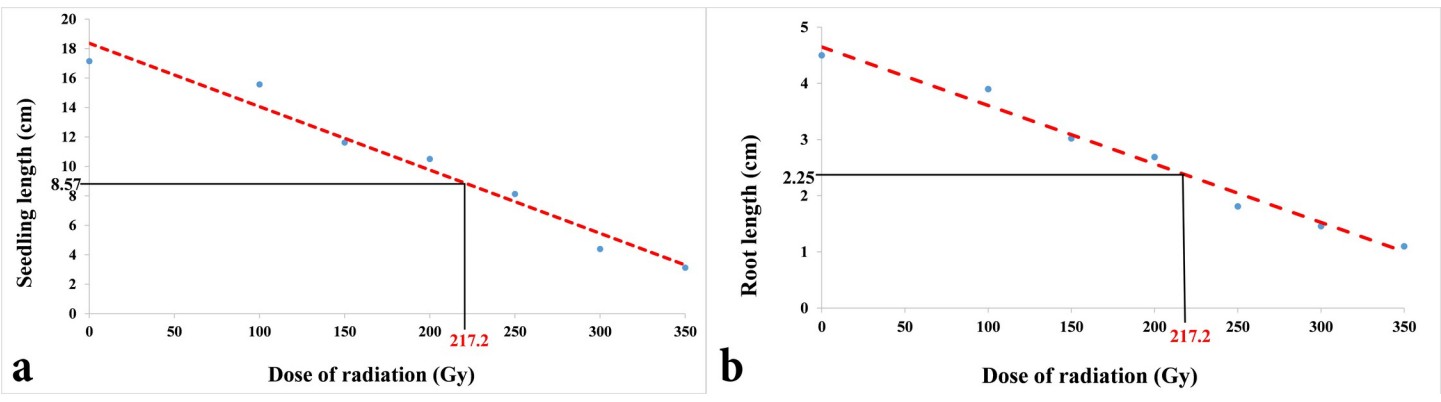

**Fig 3.** Dose response curve of (a) seedling length. (b) root length.

[33]. Moreover, it is already well established in accordance with the norms of IAEA that gamma irradiation up to the dosage of 10 kGy had no deleterious effect on human health [34]. Therefore, seeds treated with gamma rays should be considered safe when used for consumption purposes.

Probit analysis, used in the present study, is extensively utilized in research involving binomial responses. Its primary application is in toxicological investigations, where it converts the sigmoidal dose–response curve to a straight line that can then be easily evaluated using regression deploying either least squares or maximum likelihood. To simplify the complex percent affected vs. dose–response relationship into a simple linear relationship of probit vs. dose–response, 0 is defined as 0.0001, and 10 as 0.9999 and included a table to assist researchers in converting impact-percentage to probit, which is then plotted against the logarithm of dose. Moreover, this existing tool has the capability to evaluate $LD_{50}$ or $GR_{50}$ at a 95% confidence interval along with a chi-square test for checking the adequacy of fit [35].

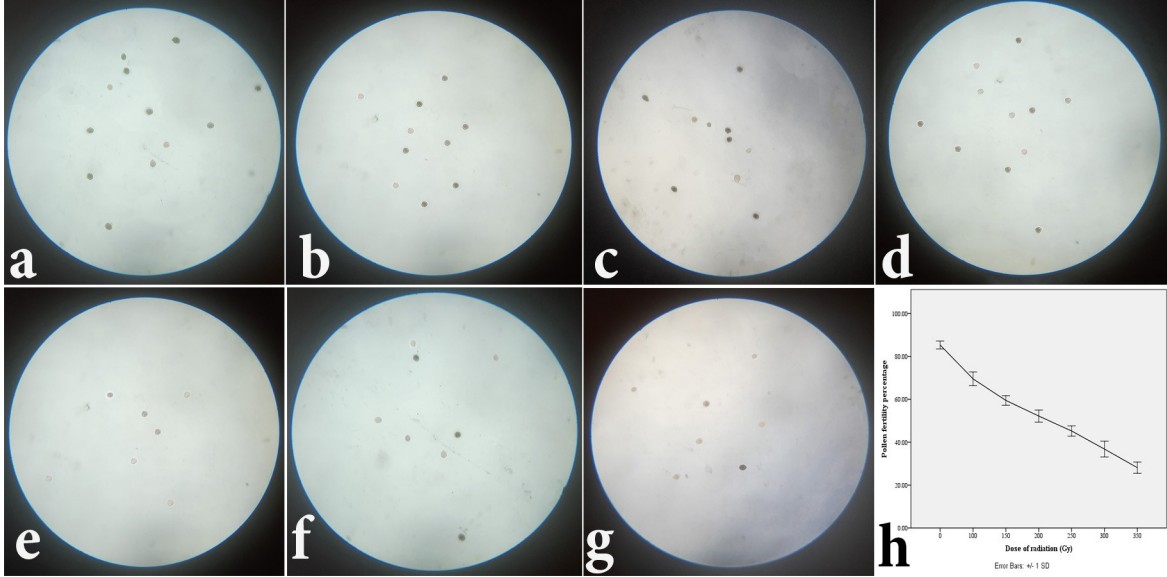

**Fig 4.** Effect of gamma irradiation on fertility percentage of pollen of lentil (*Lens culinaris* Medik.) observed under 40X opjective lens of compound light microscope (Olympus OIC); the irradiation doses are: (a) Control (0 Gy). (b) 100 Gy. (c) 150 Gy. (d) 200 Gy. (e) 250 Gy. (f) 300 Gy. (g) 350 Gy; (h) Graphical representation of the same.

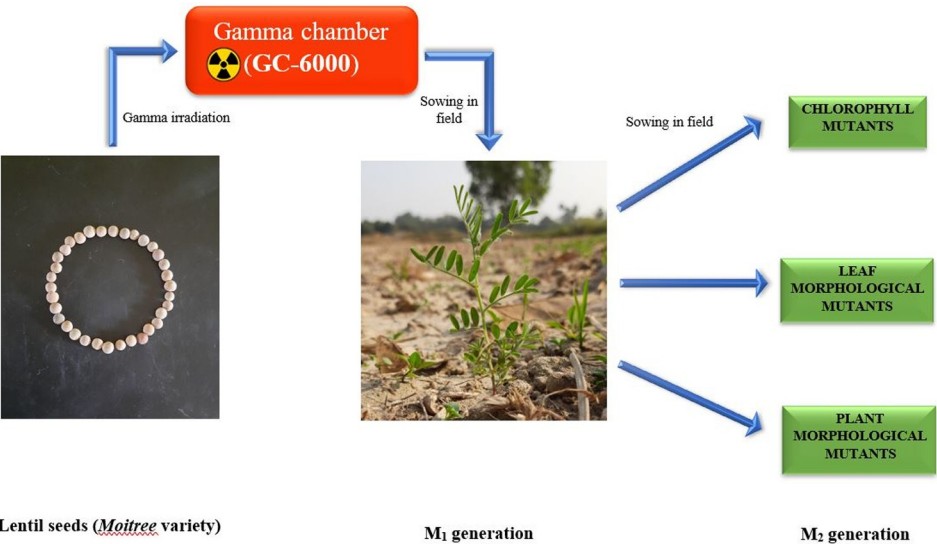

**Fig 5. Diagram of development of chloro- and morpho-mutants in M$_2$ generation.**

**Fig 6. Detection of chlorophyll and morphological mutants in M$_2$ population.** (a, g, k) Untreated plant, (b-e) Chlorophyll mutants [(b) Albina, (c) Xantha, (d) Chlorina, (e) Xantha-viridis], (f) Chimera, (h-j) Plant morphological mutants, (i-t) Leaf shape (leaf morphological) mutants, (u) Leaf arrangement in untreated plant, (v-w) Change in leaf arrangement in leaf morphological mutants.

**Table 2. Frequency of candidate mutants obtained in the $M_2$ generation.**

| Types of mutants | | Frequency of mutants obtained in specific doses of gamma radiation (%) | | | | | | Total frequency (%) |
|---|---|---|---|---|---|---|---|---|
| | | 100 Gy | 150 Gy | 200 Gy | 250 Gy | 300 Gy | 350 Gy | |
| Chlorophyll mutants | Albina | - | - | - | 0.001 | 0.001 | - | 0.002 |
| | Xantha | - | 0.002 | 0.007 | 0.01 | 0.013 | 0.008 | 0.031 |
| | Chlorina | - | 0.005 | 0.017 | 0.025 | 0.033 | 0.011 | 0.091 |
| | Xantha-viridis | - | 0.003 | 0.012 | 0.016 | 0.019 | - | 0.05 |
| | Chimera | - | - | 0.001 | 0.002 | 0.004 | - | 0.007 |
| Leaf morphological mutants | Change in leaf shape | - | 2.43 | 5.57 | 9.63 | 13.43 | 7.69 | 38.75 |
| | Change in leaf arrangement | - | 0.001 | 0.001 | 0.003 | 0.004 | - | 0.009 |
| Plant morphological mutants | | - | - | 0.007 | 0.008 | 0.011 | 0.003 | 0.029 |

The determined $GR_{50}$ value in this research, through probit analysis, was 217.2 Gy. In general, $GR_{50}$ is an effective method for quantifying a parameter's 50% decrease [23]. Because $M_1$ plants are heterozygous at the genotypic level, the change caused by mutation may be seen in any allele located in any random locus, leading to reduced mutation manifestation [36]. A lower likelihood of detecting a recessive mutation at this stage and how a larger segregating mutant population may be used to harvest unique mutations in the future were elaborated in previous studies. Additionally, the $GR_{50}$ study may be regarded as acceptable since a minimal percentage of deaths occurred compared to the $LD_{50}$ study.

Pollen fertility research has also examined the tissue-damaging properties of gamma rays. Similar to seedling height, the greater the dosage of gamma radiation, the lower the pollen fertility %. This conclusion was already endorsed by a number of studies [37–42]. Kumar, et al. [38] demonstrated that when dried seeds of pea were irradiated, a 5 kR dosage of gamma rays induced 10.06 percent pollen sterility, while a 40 kR exposure generated 53.12 percent pollen sterility. Additionally, when presoaked seeds were utilized, they observed an increase in pollen sterility. Wani [42] treated lentil seeds from the Pant L-406 and Type-8 varieties with three different chemical mutagens (EMS, sodium azide, and hydrazine hydrate). Additionally, this experiment demonstrated a decrease in pollen fertility and increased dosages of each mutagen. Notably, for each dosage, the $M_2$ generation had a higher fertility % than the $M_1$ generation [42]. Additionally, Priyanka, et al. [39], Eswaramoorthy, et al. [37], and Vikhe and Nehul [41] all found comparable findings with the mutagens they employed in horse grams (*Macrotyloma*

**Table 3. Agronomic performance of selected $M_3$ mutants: Comparison of plant height, root length, number of pods, and yield with $M_2$ generation and control.**

| Pedigree of selected mutants | Plant height (cm) | | | Root length (cm) | | | Number of pods per plant | | | Yield per plant (g) | | |
|---|---|---|---|---|---|---|---|---|---|---|---|---|
| | $M_3$ | Increment from $M_2$ generation | Increment from control | $M_3$ | Increment from $M_2$ generation | Increment from control | $M_3$ | Increment from $M_2$ generation | Increment from control | $M_3$ | Increment from $M_2$ generation | Increment from control |
| $VBM_3Sel$-34 | 39.3 | 18.5 | 26.6 | 10.9 | 4.7 | 8 | 21 | 8 | 14 | 1.67 | 0.55 | 1.11 |
| $VBM_3Sel$-57 | 27.2 | 6.4 | 14.5 | 8.3 | 2.1 | 5.4 | 23 | 10 | 16 | 1.78 | 0.66 | 1.22 |
| $VBM_3Sel$-109 | 25.8 | 5 | 13.1 | 9.5 | 3.3 | 6.6 | 19 | 6 | 12 | 1.63 | 0.51 | 1.07 |
| $VBM_3Sel$-178 | 26.7 | 5.9 | 14 | 10.2 | 4 | 7.3 | 22 | 9 | 15 | 1.7 | 0.58 | 1.14 |
| $VBM_3Sel$-259 | 25.6 | 4.8 | 12.9 | 8.6 | 2.4 | 5.7 | 16 | 3 | 9 | 1.59 | 0.47 | 1.03 |
| $VBM_3Sel$-357 | 31.3 | 10.5 | 18.6 | 12.7 | 6.5 | 9.8 | 25 | 12 | 18 | 1.99 | 0.87 | 1.43 |

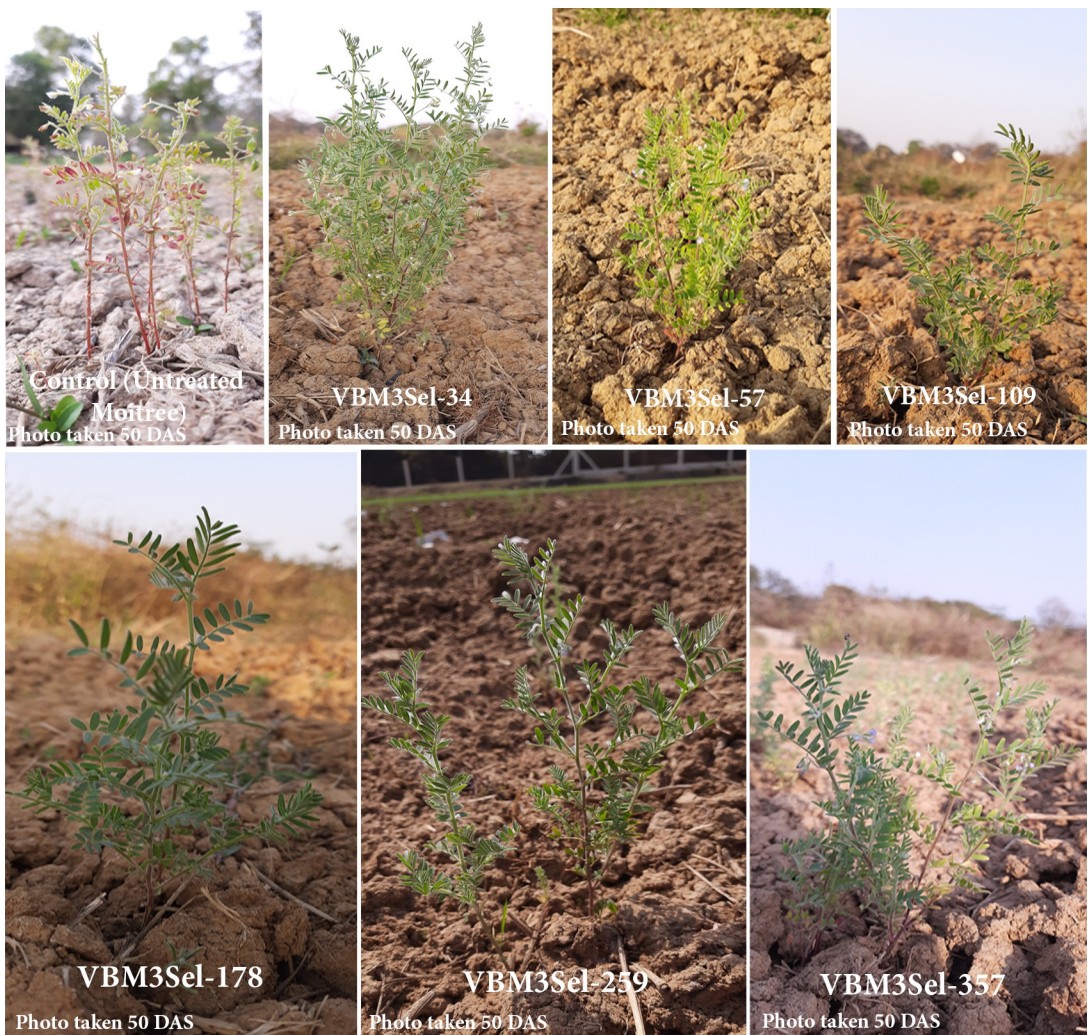

**Fig 7. Selected M₃ mutants with improved agronomic characters at 50 DAS than that of control (untreated Moitree).**

*uniflorum*), cowpea (*Vigna unguiculata* (L.) Walp. ), and mung bean (*Vigna radiata* (L.) Wilczek). On the other hand, Tamilzharasi, et al. [40] showed a reduced pollen fertility % in black gram (*Vigna mungo* (L.) Hepper) when combined higher doses of gamma rays and EMS were used rather than only gamma rays.

Albina, chlorina, xantha, and xantha-viridis chlorophyll mutants were obtained in the present investigation. Chlorina was the most prevalent, accounting for 0.091 percent of the total, followed by xantha-viridis (0.05 percent), xantha (0.031 percent), and albina (0.002 percent) (Table 2). Albina was detected only in plots with seeds that had been irradiated with 250 Gy and 300 Gy gamma rays. In the case of other chlorophyll mutants, the frequency increased up to 300 Gy but then decreased abruptly after 350 Gy gamma ray exposure. This might be due to the fatality associated with greater doses of this physical mutagen. Similarly, only plots with 200 Gy, 250 Gy, or 300 Gy showed chimera at frequencies of 0.001%, 0.002%, and 0.004%, respectively (Table 2). The highest frequency was reported when morphological mutants with altered leaf shape were detected (38.75%). Nine distinct leaf forms were identified from this group (Fig 6I–6T). Additionally, two different forms of leaf arrangement mutations (Fig 6V and 6W) and

three other types of plant morphological mutants were detected in the field (Fig 6H–6J). Furthermore, the plants treated with 300 Gy had the highest frequency of mutants, followed by the plants treated with 250 Gy.

The frequency of chlorophyll mutation is an excellent indicator of various mutagen dosages [43]. Additionally, chlorophyll abnormalities generated by irradiation may be utilized to locate accessible mutations in the irradiated population. Ionizing radiation has the potential to modify the photosynthetic complex [44] by decreasing the photosystem's efficiency [45]. A substantial dosage of gamma rays, up to 500 Gy, lowers the chlorophyll content by 80.91% and the grana and stroma thylakoid organization patterns [46]. Apart from those fundamental outcomes, stable mutants with good agronomic performances can directly be used as breeding materials for making novel genotypes [47, 48].

It was observed that the selected $M_3$ generation mutants exhibited significant improvements in agronomic traits compared to their respective $M_2$ mean and control values. This suggests that induced mutations have effectively led to the development of desirable genetic variations, which can be exploited for future crop improvement efforts. These findings align with previous studies on mutation breeding, which have successfully generated mutants with superior agronomic traits in various crop species [27, 28]. So, the present study underscores the potential of gamma-ray-induced mutagenesis in developing lentil mutants with improved agronomic characteristics. The selected $M_3$ generation mutants can serve as valuable genetic resources for breeders to incorporate into lentil breeding programs, ultimately contributing to enhanced crop performance and yield [49]. Further studies involving advanced generations ($M_4$ and beyond) and molecular characterization of the induced mutants are recommended to better understand the genetic basis of the observed phenotypic changes and confirm the traits' stability across generations.

## Conclusions

In conclusion, this study successfully demonstrated the efficacy of gamma-ray-induced mutagenesis in lentil for the development of novel variants with improved agronomic traits. The optimal dosage of gamma radiation was determined based on the $GR_{50}$ value and seedling parameters. Distinct types of chlorophyll mutants were identified in the $M_2$ generation, along with several leaf and plant morphological mutants. Furthermore, six promising mutants were selected in the $M_3$ generation, exhibiting superior agronomic traits such as increased plant height, root length, number of pods, and yield per plant. These findings serve as a valuable resource for lentil breeding programs to enhance the crop's performance and productivity. Future research should focus on conducting multi-environment trials to evaluate the stability and adaptability of these selected mutants across diverse agroecological zones. Additionally, further genetic and molecular characterization of these mutants will help to understand the underlying mechanisms involved in the expression of the improved traits, thereby facilitating the development of molecular markers for marker-assisted selection. Lastly, it would be essential to involve farmers in participatory varietal selection to ensure the acceptability and adoption of these improved lentil varieties at the grassroots level.

## Acknowledgments

The research article is a part of Mr. Biswajit Pramanik for his Ph.D. research program, which was supervised by Dr. Sandip Debnath at Visva-Bharati University. Researchers also acknowledge the help received from the YSRA-BRNS project entitled "Physical mutation to improve the P acquisition along with nodulation efficiency in lentil", (No. 55/14/17/2020-BRNS/10360 Dated 10/12/2020) from the Department of Atomic Energy, BARC, Mumbai, India.

## Author Contributions

**Conceptualization:** Biswajit Pramanik, Sandip Debnath.

**Data curation:** Biswajit Pramanik.

**Formal analysis:** Biswajit Pramanik.

**Investigation:** Sandip Debnath.

**Methodology:** Sandip Debnath.

**Project administration:** Sandip Debnath.

**Software:** Mehdi Rahimi.

**Supervision:** Sandip Debnath.

**Validation:** Sandip Debnath.

**Writing – original draft:** Biswajit Pramanik, Sandip Debnath, Mehdi Rahimi, Md. Mostofa Uddin Helal, Rakibul Hasan.

**Writing – review & editing:** Biswajit Pramanik, Sandip Debnath, Mehdi Rahimi, Md. Mostofa Uddin Helal, Rakibul Hasan.

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
