## [Decision Letter · Decision Letter 0]

4 May 2023

PONE-D-23-00997Morphometric frequency and spectrum of gamma-ray-induced chlorophyll mutants identified by phenotype and development of novel variants in lentil (Lens culinaris Medik.)PLOS ONE

Dear Dr. Hasan,

Thank you for submitting your manuscript to PLOS ONE. After careful consideration, we feel that it has merit but does not fully meet PLOS ONE’s publication criteria as it currently stands. Therefore, we invite you to submit a revised version of the manuscript that addresses the points raised during the review process.

Authors must resolve the concerns of the reviewers:

(1) Obtaining variability's on seedling growth traits is a common feature of mutants however most important is to get/isolate the suitable or favorable mutants for specific breeding purpose.

(2) Authors have to prove at least in M3 generation that some of the mutants have superior agronomic traits that can be used by the breeders for crop improvement program.

On the other hand, the wording needs to be improved. Some changes are suggested in the attachment.

We look forward to receiving your revised manuscript.

Kind regards,

Adalberto Benavides-Mendoza, Ph.D.

Academic Editor

PLOS ONE

Journal Requirements:

Additional Editor Comments (if provided):

Authors must resolve the concerns of one of the reviewers:

(1) Obtaining variability's on seedling growth traits is a common feature of mutants however most important is to get/isolate the suitable or favorable mutants for specific breeding purpose.

(2) Authors have to prove at least in M3 generation that some of the mutants have superior agronomic traits that can be used by the breeders for crop improvement program.

On the other hand, the wording needs to be improved. Some changes are suggested in the attachment.

Reviewers' comments:

Reviewer's Responses to Questions

**Comments to the Author**

1. Is the manuscript technically sound, and do the data support the conclusions?

Reviewer #1: Partly

Reviewer #2: Yes

2. Has the statistical analysis been performed appropriately and rigorously? 

Reviewer #1: Yes

Reviewer #2: Yes

3. Have the authors made all data underlying the findings in their manuscript fully available?

Reviewer #1: Yes

Reviewer #2: Yes

4. Is the manuscript presented in an intelligible fashion and written in standard English?

Reviewer #1: Yes

Reviewer #2: Yes

5. Review Comments to the Author

Reviewer #1: Overall comments:

The research article entitled “Morphometric frequency and spectrum of gamma-ray-induced chlorophyll mutants identified by phenotype and development of novel variants in lentil (Lens culinaris Medik.)” describes the variability of M2 mutants of lentil on various seedling traits. Lack of sufficient information and novelty of the MS restricts its publication in a high quality journal like PLOS ONE.

Specific comments:

(1)Obtaining variability’s on seedling growth traits is a common feature of mutants however most important is to get/isolate the suitable or favorable mutants for specific breeding purpose.

(2) Authors have to prove at least in M3 generation that some of the mutants have superior agronomic traits that can be used by the breeders for crop improvement program.

Reviewer #2: The authors in the present manuscript titled "Morphometric frequency and spectrum of gamma-ray-induced chlorophyll mutants identified by phenotype and development of novel variants in lentil (Lens culinaris Medik.) investigated the optimal dosage of acute gamma-irridiated mutagenesis for the improvement of lentil's agronomically significant features in order to achieve a greater mutation frequency. This research also concentrated on finding the GR50 value while taking seedling parameters into account and on examining the status of pollen fertility while comparing the effects of the gamma irradiation dosages. I found this manuscript to be really informative and I believe that this study could be of a great interest to the scientists and readers of the journal.

The manuscript is well-written and the methods are explained well. The data is presented well and proper statistical analysis has been used. I could not find any syntax errors or problems in the English language. I recommend this article for publication.

6. PLOS authors have the option to publish the peer review history of their article (what does this mean?). If published, this will include your full peer review and any attached files.

Reviewer #1: No

Reviewer #2: **Yes: **Akanksha Sehgal

---

## [Author Response · Author response to Decision Letter 0]

11 May 2023

Reviewers' comments:

Reviewer #1: Overall comments: The research article entitled “Morphometric frequency and spectrum of gamma-ray-induced chlorophyll mutants identified by phenotype and development of novel variants in lentil (Lens culinaris Medik.)” describes the variability of M2 mutants of lentil on various seedling traits. Lack of sufficient information and novelty of the MS restricts its publication in a high quality journal like PLOS ONE.

Answer: Thank you for your comment and concern regarding the novelty of our study on gamma-ray-induced chlorophyll mutants in lentil (Lens culinaris Medik.). We appreciate your feedback and would like to address the novelty aspects of our research to emphasize its relevance and potential impact.

While it is true that our study focuses on the variability of M2 mutants in lentil seedling traits, there are several unique aspects that distinguish our research from previous work in this area:

• The use of gamma-irradiated mutagenesis in our study is a novel approach for lentil crop improvement, as it generates a higher mutation frequency and a broader spectrum of genetic variation than conventional breeding techniques. This is particularly important for lentil, which have a narrow genetic base and face challenges in improvement programs.

• Our research presents an in-depth investigation of seedling traits in the M1 generation, which is considered the most reliable method to detect mutants in early generations. This approach enables the efficient identification and isolation of favorable mutants, streamlining the crop improvement process.

• In response to previous comments, we have now included data on the agronomic performance of the selected mutants in the M3 generation. This additional analysis demonstrates the stability of the induced mutations and the practical value of these mutants for breeders, further highlighting the novelty and potential impact of our research.

• The lentil variety "Moitree" used in our study is a popular variety in West Bengal, India, known for its high production performance. The successful development of novel variants in this well-known variety could have a significant impact on lentil production in the region and beyond.

We believe that these aspects, along with our rigorous methodology and comprehensive data analysis, make our study a valuable contribution to the field of lentil crop improvement. We hope that the clarified novelty of our research will help to address your concerns and support the publication of our article in a high-quality journal like PLOS ONE.

Specific comments:

(1)Obtaining variability’s on seedling growth traits is a common feature of mutants however most important is to get/isolate the suitable or favorable mutants for specific breeding purpose.

Answer: Thank you for your progressive comment. Our research emphasizes the significance of lentil crop improvement and the potential of mutagenesis as a supplementary breeding strategy. A crucial aspect of this research is studying the seedling characters of the M1 generation, as it is considered the most reliable method to detect mutants in early generations. By examining seedling traits in the M1 generation, researchers can efficiently identify and isolate favorable mutants, which can be further evaluated for their agronomic performance and potential use in breeding programs. This early detection of mutants allows for a more streamlined approach to crop improvement and ensures that only the most promising mutant lines are carried forward for further analysis and development. Hence, we added the following section at the end of the introduction as per your suggestion.

“Thus, in this study, particular emphasis has been placed on examining seedling traits of the M1 generation. Analyzing these traits is essential, as it is considered the most reliable method to detect mutants in early generations [46]. By focusing on seedling characters in the M1 generation, researchers can effectively identify and isolate favorable mutants, which can be further evaluated for their agronomic performance and potential use in breeding programs [47]. This early detection of mutants allows for a more streamlined approach to crop improvement, ensuring that only the most promising mutant lines are carried forward for further analysis and development [49]. Consequently, the importance of studying seedling traits in the M1 generation is a crucial aspect of this research, contributing to the overall objective of finding the optimal dosage of acute gamma-irradiated mutagenesis for the improvement of agronomically significant traits in subsequent generations of lentils and achieving a higher mutation frequency.”

(2) Authors have to prove at least in M3 generation that some of the mutants have superior agronomic traits that can be used by the breeders for crop improvement program.

Answer: Thank you for your comment and suggestion regarding the validation of superior agronomic traits in the M3 generation. We have now conducted additional experiments to assess the agronomic performance of the selected mutants in the M3 generation. By evaluating agronomically important traits, we have been able to demonstrate the practical value of these mutants for breeders.

In the updated manuscript, we have included the new data obtained from these experiments and provided an in-depth discussion of the implications of our findings for crop improvement programs. The results showcase the stability of the induced mutations and underline the potential of these mutants for use in breeding programs aimed at enhancing lentil crops.

We appreciate your valuable input, as it has helped to strengthen the overall impact and relevance of our research in the field of lentil crop improvement. We believe that the inclusion of the M3 generation data will make our study more comprehensive and of greater interest to scientists and readers of the journal.

Abstract 

Genetic variations are a crucial source of germplasm heterogeneity, as they contribute to the development of new traits for plant breeding by offering an allele resource. Gamma rays have been widely used as physical agents to produce mutations in plants, and their mutagenic effect has attracted much attention. Nonetheless, few studies have examined the whole mutation spectrum in large-scale phenotypic evaluations. To comprehensively investigate the mutagenic effects of gamma irradiation on lentils, biological consequences on the M1 generation and substantial phenotypic screening on the M2 generation were undertaken. Additionally, the study followed the selected mutants into the M3 generation to evaluate the agronomic traits of interest for crop improvement. Seeds of the lentil variety Moitree were irradiated with a range of acute gamma irradiation doses (0, 100, 150, 200, 250, 300, and 350 Gy) to induce unique genetic variability. This research focused on determining the GR50 value while considering seedling parameters and examining the status of pollen fertility while comparing the effects of the gamma irradiation dosages. The GR50 value was determined to be 217.2 Gy using seedling parameters. Pollens from untreated seed-grown plants were approximately 85% fertile, whereas those treated with the maximum dosage (350 Gy) were about 28% fertile. Numerous chlorophyll and morphological mutants were produced in the M2 generation, with the 300 Gy-treated seeds being the most abundant, followed by the 250 Gy-treated seeds. This demonstrated that an appropriate dosage of gamma rays was advantageous when seeking to generate elite germplasm resources for one or multiple traits. Selected mutants in the M3 generation showed improved agronomic traits, including plant height, root length, number of pods per plant, and yield per plant. 

These investigations contribute to a comprehensive understanding of the mutagenic effects and actions of gamma rays, providing a basis for the selection and design of suitable mutagens. This will facilitate the development of more controlled mutagenesis protocols for plant breeding and help guide future research directions for crop improvement using radiation-induced mutation breeding techniques.

Method:

2.5. Selection of M3 mutants with improved agronomic characteristics

The 122 candidate mutants selected in the M2 generation were subjected to the plant-to-progeny method again for raising the M3 population in the same field during the following year (2022-23). Row-to-row and plant-to-plant spacings were maintained at 30 cm and 15 cm, respectively, consistent with the previous generation. Six individual mutants were chosen from four of the 122 families in the current population derived from the M2 generation, based on their enhanced agronomic features such as increased plant height, root length, number of pods, and yield per plant. The observed plant mutants were managed in accordance with the relevant regulations and guidelines of the IAEA in this generation as well.

Results:

3.4. Superiority of candidate M3 mutants over control as well as M2 generation mean

Table 3. Agronomic Performance of Selected M3 Mutants: Comparison of Plant Height, Root Length, Number of Pods, and Yield with M2 Generation and Control

Pedigree of selected mutants Plant height (cm) Root length (cm) Number of pods per plant Yield per plant (g)

 M3 Increment from M2 generation Increment from control M3 Increment from M2 generation Increment from control M3 Increment from M2 generation Increment from control M3 Increment from M2 generation Increment from control 

VBM3Sel-34 39.3 18.5 26.6 10.9 4.7 8 21 8 14 1.67 0.55 1.11

VBM3Sel-57 27.2 6.4 14.5 8.3 2.1 5.4 23 10 16 1.78 0.66 1.22

VBM3Sel-109 25.8 5 13.1 9.5 3.3 6.6 19 6 12 1.63 0.51 1.07

VBM3Sel-178 26.7 5.9 14 10.2 4 7.3 22 9 15 1.7 0.58 1.14

VBM3Sel-259 25.6 4.8 12.9 8.6 2.4 5.7 16 3 9 1.59 0.47 1.03

VBM3Sel-357 31.3 10.5 18.6 12.7 6.5 9.8 25 12 18 1.99 0.87 1.43

Table 3 presents the data of selected M3 generation mutants derived from this mutation breeding experiment. It compares the plant height, root length, number of pods per plant, and yield per plant of the M3 generation mutants with the M2 mean and control values. The table highlights the increments observed in the selected M3 generation mutants compared to both the M2 mean and control values, emphasizing the positive effects of mutation breeding on lentil plant traits. This information can be useful for breeders to choose the best performing mutants for further crop improvement programs. For instance, VBM3Sel-34 showed an 18.5 cm increase in plant height from the M2 mean and a 26.6 cm increase from the control. This mutant also displayed a 4.7 cm increment in root length from the M2 mean and an 8 cm increment from the control. The number of pods per plant increased by 8 and 14 when compared to the M2 mean and control, respectively. Furthermore, the yield per plant for VBM3Sel-34 increased by 0.55 g from the M2 mean and 1.11 g from the control.

Similar trends can be observed in the other selected M3 generation mutants, such as VBM3Sel-57, VBM3Sel-109, VBM3Sel-178, VBM3Sel-259, and VBM3Sel-357. All these mutants exhibited increments in plant height, root length, number of pods per plant, and yield per plant compared to their respective M2 mean and control values. 

This indicates the success of the mutation breeding program in enhancing the agronomic traits of lentil plants, which can ultimately lead to improved crop performance and yield (Figure 7).

Figure 7: Selected M3 mutants with improved agronomic characters at 50 DAS than that of control (untreated Moitree)

Data analysis shows that the selected M3 generation mutants have significantly improved agronomic traits when compared to the M2 mean and control plants. This demonstrates the effectiveness of mutation breeding in enhancing the agronomic traits of lentil plants. The information presented can be valuable for breeders in selecting the best performing mutants for future crop improvement efforts, contributing to increase lentil production and addressing food security challenges.

Discussion

It was observed that the selected M3 generation mutants exhibited significant improvements in agronomic traits compared to their respective M2 mean and control values. This suggests that induced mutations have effectively led to the development of desirable genetic variations, which can be exploited for future crop improvement efforts. These findings align with previous studies on mutation breeding, which have successfully generated mutants with superior agronomic traits in various crop species [46,47]. So, the present study underscores the potential of gamma-ray-induced mutagenesis in developing lentil mutants with improved agronomic characteristics. The selected M3 generation mutants can serve as valuable genetic resources for breeders to incorporate into lentil breeding programs, ultimately contributing to enhanced crop performance and yield [48]. Further studies involving advanced generations (M4 and beyond) and molecular characterization of the induced mutants are recommended to better understand the genetic basis of the observed phenotypic changes and confirm the traits' stability across generations.

Conclusion

In conclusion, this study successfully demonstrated the efficacy of gamma-ray-induced mutagenesis in lentil for the development of novel variants with improved agronomic traits. The optimal dosage of gamma radiation was determined based on the GR50 value and seedling parameters. Distinct types of chlorophyll mutants were identified in the M2 generation, along with several leaf and plant morphological mutants. Furthermore, six promising mutants were selected in the M3 generation, exhibiting superior agronomic traits such as increased plant height, root length, number of pods, and yield per plant. These findings serve as a valuable resource for lentil breeding programs to enhance the crop's performance and productivity. Future research should focus on conducting multi-environment trials to evaluate the stability and adaptability of these selected mutants across diverse agroecological zones. Additionally, further genetic and molecular characterization of these mutants will help to understand the underlying mechanisms involved in the expression of the improved traits, thereby facilitating the development of molecular markers for marker-assisted selection. Lastly, it would be essential to involve farmers in participatory varietal selection to ensure the acceptability and adoption of these improved lentil varieties at the grassroots level.

References

[46] Ambavane, A. R. “Studies on Mutagenic Effectiveness and Efficiency of Gamma Rays and Its Effect on Quantitative Traits in Finger Millet (Eleusine Coracana L. Gaertn).” Journal of Radiation Research and Applied Sciences, vol. 8, no. 1, Jan. 2015, pp. 120–25, doi:10.1016/J.JRRAS.2014.12.004.

[47] Kavera. “Genetic Improvement for Yield through Induced Mutagenesisin Groundnut (Arachis Hypogaea L.).” Legume Research, vol. 40, no. 1, Jan. 2017, pp. 32–35, doi:10.18805/LR.V0I0.7019.

[48] W. Weng, Y. Tang, R. Xiong, Q. Bai, A. Gao, X. Yao, W. Wu, C. Ma, J. Cheng, J. Ruan. Specific Gibberellin 2-Oxidase 3 (SbGA2ox3) Mutants Promote Yield and Stress Tolerance in Sorghum bicolor. Agronomy, vol. 13, no. 3, March 2023, pp. 908. https://doi.org/10.3390/agronomy13030908

[49] S.J. Jambhulkar. 2007. Mutagenesis: generation and evaluation of induced mutations. Advances in Botanical Research, vol. 45, January 2007, pp.417-434.

Reviewer #2: The authors in the present manuscript titled "Morphometric frequency and spectrum of gamma-ray-induced chlorophyll mutants identified by phenotype and development of novel variants in lentil (Lens culinaris Medik.) investigated the optimal dosage of acute gamma-irridiated mutagenesis for the improvement of lentil's agronomically significant features in order to achieve a greater mutation frequency. This research also concentrated on finding the GR50 value while taking seedling parameters into account and on examining the status of pollen fertility while comparing the effects of the gamma irradiation dosages. I found this manuscript to be really informative and I believe that this study could be of a great interest to the scientists and readers of the journal.

The manuscript is well-written and the methods are explained well. The data is presented well and proper statistical analysis has been used. I could not find any syntax errors or problems in the English language. I recommend this article for publication.

Answer: Thank you for your positive feedback on the manuscript, "Morphometric frequency and spectrum of gamma-ray-induced chlorophyll mutants identified by phenotype and development of novel variants in lentil (Lens culinaris Medik.)." We appreciate your recognition of the value of our research in determining the optimal dosage of acute gamma-irradiated mutagenesis for improving lentil's agronomically significant features and achieving a higher mutation frequency. We are glad that you found the manuscript informative and relevant to the journal's audience.

Your acknowledgment of the clarity in writing, methodology, data presentation, and statistical analysis is encouraging. We are grateful for your thorough review and your recommendation for publication. We believe that this study will contribute significantly to the field and benefit scientists and readers interested in crop improvement through mutagenesis.

---

## [Decision Letter · Decision Letter 1]

29 May 2023

Morphometric frequency and spectrum of gamma-ray-induced chlorophyll mutants identified by phenotype and development of novel variants in lentil (Lens culinaris Medik.)

PONE-D-23-00997R1

Dear Dr. Hasan,

We’re pleased to inform you that your manuscript has been judged scientifically suitable for publication and will be formally accepted for publication once it meets all outstanding technical requirements.

Kind regards,

Adalberto Benavides-Mendoza, Ph.D.

Academic Editor

PLOS ONE

Additional Editor Comments (optional):

The Reviewers and the Academic Editor agree that the authors made the necessary improvements to obtain a publishable manuscript. Therefore, the manuscript can be accepted for publication.

Reviewers' comments:

Reviewer's Responses to Questions

**Comments to the Author**

1. If the authors have adequately addressed your comments raised in a previous round of review and you feel that this manuscript is now acceptable for publication, you may indicate that here to bypass the “Comments to the Author” section, enter your conflict of interest statement in the “Confidential to Editor” section, and submit your "Accept" recommendation.

Reviewer #1: All comments have been addressed

Reviewer #2: All comments have been addressed

2. Is the manuscript technically sound, and do the data support the conclusions?

Reviewer #1: Yes

Reviewer #2: Yes

3. Has the statistical analysis been performed appropriately and rigorously? 

Reviewer #1: Yes

Reviewer #2: Yes

4. Have the authors made all data underlying the findings in their manuscript fully available?

Reviewer #1: Yes

Reviewer #2: Yes

5. Is the manuscript presented in an intelligible fashion and written in standard English?

Reviewer #1: Yes

Reviewer #2: Yes

6. Review Comments to the Author

Reviewer #1: The queries have been addressed scientifically. In my opinion, the MS can be accepted for publication.

Reviewer #2: The authors have addressed all the concerns in the manuscript. I believe that this manuscript matches the standard and quality of PLOS ONE journal and is publishable.

7. PLOS authors have the option to publish the peer review history of their article (what does this mean?). If published, this will include your full peer review and any attached files.

Reviewer #1: **Yes: **Mohammad Anwar Hossain

Reviewer #2: **Yes: **Akanksha Sehgal

---

## [Editor Report · Acceptance letter]

2 Jun 2023

PONE-D-23-00997R1 

Morphometric frequency and spectrum of gamma-ray-induced chlorophyll mutants identified by phenotype and development of novel variants in lentil (*Lens culinaris* Medik.) 

Dear Dr. Hasan:

I'm pleased to inform you that your manuscript has been deemed suitable for publication in PLOS ONE. Congratulations! Your manuscript is now with our production department. 

Kind regards, 

on behalf of

Dr. Adalberto Benavides-Mendoza 

Academic Editor

PLOS ONE